# Predicting Rheumatoid Arthritis Development Using Hand Ultrasound and Machine Learning—A Two-Year Follow-Up Cohort Study

**DOI:** 10.3390/diagnostics14111181

**Published:** 2024-06-04

**Authors:** Mahyar Daskareh, Azin Vakilpour, Erfan Barzegar-Golmoghani, Saeid Esmaeilian, Samira Gilanchi, Fatemeh Ezzati, Majid Alikhani, Elham Rahmanipour, Niloofar Amini, Mohammad Ghorbani, Parham Pezeshk

**Affiliations:** 1Department of Radiology, University of California San Diego, San Diego, CA 92093, USA; mdaskareh@health.ucsd.edu; 2Division of Cardiovascular Diseases, Department of Medicine, Hospital of the University of Pennsylvania, Philadelphia, PA 19104, USA; azinvkp@gmail.com; 3Department of Biomedical Engineering, Tarbiat Modares University, Tehran 14115-111, Iran; erfanbarzegar92@gmail.com; 4Department of Radiology, Shiraz University of Medical Sciences, Shiraz 71348, Iran; kmssd87@gmail.com; 5Proteomics Research Center, Shahid Beheshti University of Medical Sciences, Tehran 19839-63113, Iran; samira.guilanchi@gmail.com; 6Division of Rheumatic Disease, Department of Internal Medicine, UT Southwestern Medical Center, Dallas, TX 75390, USA; fatemeh.ezzati@utsouthwestern.edu; 7Department of Internal Medicine, Rheumatology Research Center, Shariati Hospital, Tehran University of Medical Sciences, Tehran 14117-13135, Iran; nilooamini94@gmail.com; 8Immunology Research Center, Mashhad University of Medical Sciences, Mashhad 91779-48564, Iran; elhamrahmanipour@gmail.com; 9Orthopedic Research Center, Mashhad University of Medical Sciences, Mashhad 91779-48564, Iran; m.ghorbani96@gmail.com; 10Division of Musculoskeletal Imaging, Department of Radiology, UT Southwestern Medical Center, Dallas, TX 75390, USA

**Keywords:** arthritis, rheumatoid, synovitis, ultrasonography, machine learning

## Abstract

Background: The early diagnosis and treatment of rheumatoid arthritis (RA) are essential to prevent joint damage and enhance patient outcomes. Diagnosing RA in its early stages is challenging due to the nonspecific and variable clinical signs and symptoms. Our study aimed to identify the most predictive features of hand ultrasound (US) for RA development and assess the performance of machine learning models in diagnosing preclinical RA. Methods: We conducted a prospective cohort study with 326 adults who had experienced hand joint pain for less than 12 months and no clinical arthritis. We assessed the participants clinically and via hand US at baseline and followed them for 24 months. Clinical progression to RA was defined according to the ACR/EULAR criteria. Regression modeling and machine learning approaches were used to analyze the predictive US features. Results: Of the 326 participants (45.10 ± 11.37 years/83% female), 123 (37.7%) developed clinical RA during follow-up. At baseline, 84.6% of the progressors had US synovitis, whereas 16.3% of the non-progressors did (*p* < 0.0001). Only 5.7% of the progressors had positive PD. Multivariate analysis revealed that the radiocarpal synovial thickness (OR = 39.8), PIP/MCP synovitis (OR = 68 and 39), and wrist effusion (OR = 12.56) on US significantly increased the odds of developing RA. ML confirmed these US features, along with the RF and anti-CCP levels, as the most important predictors of RA. Conclusions: Hand US can identify preclinical synovitis and determine the RA risk. The radiocarpal synovial thickness, PIP/MCP synovitis, wrist effusion, and RF and anti-CCP levels are associated with RA development.

## 1. Introduction

Rheumatoid arthritis (RA) is a chronic inflammatory condition characterized by joint inflammation and progressive joint damage, primarily affecting the hands and feet [1,2]. The timely diagnosis and effective management of RA are critical to prevent irreversible joint destruction and functional impairment [3,4,5,6]. However, early detection poses a challenge due to the often subtle and nonspecific initial symptoms [7,8]. There is a pressing need for accurate diagnostic methods capable of identifying RA in its early stages, particularly among individuals with undifferentiated arthritis [9,10,11,12].

Imaging modalities such as ultrasound (US) [13,14] and magnetic resonance imaging (MRI) have emerged as valuable tools for the detection of early inflammatory changes in the body, even before the onset of clinical synovitis [15,16]. Imaging modalities such as US and MRI [17] have emerged as valuable tools for detecting early inflammatory changes before the onset of clinical synovitis [18,19,20]. High-frequency ultrasound has shown promise as a cost-effective, non-invasive, and widely available method for the detection of mild synovitis and in predicting the progression to clinical arthritis [9,19,21]. However, previous research on the effectiveness of US in early disease detection has been limited by small sample sizes and variability [17,22,23,24].

This study aimed to evaluate the prognostic utility of hand ultrasound (US) in the early diagnosis of RA among 326 individuals presenting with nontraumatic hand joint pain who initially did not meet the established criteria for RA. Leveraging supervised machine learning techniques, we analyzed the US findings alongside other clinical and serological data to identify predictive features for RA development over an extended observation period. The insights garnered from this research have the potential to inform the development of diagnostic and prognostic models integrating ultrasound assessment for risk stratification in individuals with undifferentiated arthritis. The early identification of high-risk individuals enables the targeted implementation of disease-modifying treatments, thereby optimizing patient outcomes.

## 2. Materials and Methods

### 2.1. Study Design

We carried out a single-center, prospective cohort study at the rheumatology clinic of Dr. Shariati Hospital in Tehran, Iran, from 2018 to 2021, to assess the diagnostic effectiveness of hand ultrasonography in the early detection of RA among patients with nontraumatic hand joint pain. The primary goal was to evaluate how well ultrasound imaging could identify patients who would develop clinically evident RA within a two-year follow-up period. A power analysis, assuming a 20% difference in RA incidence between the exposed and unexposed groups, with a significance level of 0.05 and 80% power, guided our target enrollment of 300 patients.

### 2.2. Participants and Clinical Assessment

Eligible participants were individuals aged 18 years or older, presenting to the clinic with nontraumatic hand joint pain with a duration of less than 12 months and no prior diagnosis of rheumatic disease [25]. Exclusion criteria included symptoms attributable solely to degenerative joint disease, a definite RA diagnosis at initial presentation, previous arthritis, treatment with disease-modifying antirheumatic drugs (DMARDs) or glucocorticoids, systemic infections or autoimmune diseases, lymphoproliferative disorders, a history of cancer or recent radiotherapy, and refusal to participate or attend follow-up visits.

Participants provided written informed consent; completed a baseline questionnaire detailing their symptoms and family and medical history; and underwent clinical examinations and laboratory tests. Tests included the erythrocyte sedimentation rate (ESR), C-reactive protein (CRP), IgM-RF, and anti-cyclic citrullinated peptide (anti-CCP) antibodies, using standardized equipment (HITACHI 912 autoanalyzer; Stat Fax 4200 ELISA Microplate Reader, Tokyo, Japan), as per the manufacturer guidelines.

The primary outcome measure, the development of rheumatoid arthritis (RA) within a 24-month period, was determined using the 2010 American College of Rheumatology/European League Against Rheumatism (ACR/EULAR) classification criteria. The evaluation of RA development was performed by a rheumatologist who was blinded to the patients’ ultrasonography results [25]. Detailed information about diagnostic criteria and abbreviations can be found in the Appendix A.

### 2.3. Ultrasonography Examination

All patients underwent hand ultrasonography within one week of clinical assessment, performed by a radiologist with over four years of experience in musculoskeletal ultrasonography, who was unaware of the patients’ clinical and laboratory data. Using a MyLabX8 Platform (Esaote, Genoa, Italy) with a multifrequency (LA523, 2–9 MHz) linear array probe, the radiologist scanned various hand joints, including the bilateral proximal interphalangeal (PIP), metacarpophalangeal (MCP), and distal interphalangeal (DIP) joints; the interphalangeal joint (IP) of the thumb; the first carpometacarpal (CMC) joint; and the wrist joints. Each scan lasted about 20 min, following a standardized OMERACT protocol, involving a midline 12 o’clock longitudinal scan perpendicular to the joint surface [5,26].

The radiologist utilized both grayscale (GS) and power Doppler (PD) imaging to evaluate synovitis, effusion, erosion, and cartilage damage, applying a semiquantitative scoring system (0–3) [25,26]. Synovitis was identified by a GS grade above 1 or any PD activity. The radiologist recorded the GS and PD scores, along with each patient’s exposure status, in a digital database. The PD settings were optimized for sensitivity and artifact reduction, with adjustments made to the pulse repetition frequency and the positioning of the color box to accurately capture blood flow signals.

The ultrasonography equipment was regularly calibrated to ensure accuracy, and patient images were stored on a local PACS server. After data collection, all images were reviewed collectively to discuss the findings (Figure 1, Figure 2, Figure 3 and Figure 4).

### 2.4. Follow-Up

We monitored the patients for two years after the initial assessment. We conducted phone interviews with patients who did not wish to visit the clinic for their 6-, 12-, and 18-month evaluations. We inquired about their symptoms, treatments, and outcomes and advised them to visit the outpatient clinic for further examination by a rheumatologist if they showed signs of clinical synovitis. At the 24-month visit, we determined if the patients had rheumatoid arthritis or non-inflammatory pain based on the rheumatologist’s decision using the 2010 ACR/EULAR criteria (score of more than 6/10) [25].

### 2.5. Data Preprocessing and Model Development

The GS and PD scores for each assessed joint were used directly as input features for the machine learning models, without additional preprocessing. We employed several supervised learning algorithms, including decision trees, support vector machines (linear, quadratic, and Gaussian kernel), k-nearest neighbors, AdaBoost, neural networks, and random forests, to identify ultrasound characteristics predictive of preclinical RA.

The selection of the final model was guided by the documented performance in medical image classification tasks. After careful consideration, the random forest (RF) algorithm, an ensemble method that amalgamates multiple decision trees to enhance the prediction accuracy and mitigate overfitting, was deemed most suitable [27,28,29,30].

The hyperparameters, including the number of trees (set to 80), were fine-tuned using a grid search approach to optimize the model performance. The performance of the RF model was assessed using k-fold cross-validation (k = 5) and a comprehensive set of metrics, including sensitivity, specificity, precision, the F1-score, and the Matthews correlation coefficient (MCC). This approach ensured robust evaluation, particularly considering the class imbalance present in the dataset [31].

### 2.6. Statistical Analysis

Statistical analysis was performed using SPSS version 22. Quantitative data were presented as means and standard deviations, while qualitative data were expressed as frequencies and percentages. We assessed group differences using independent-samples *t*-tests for continuous data and chi-square or Fisher’s exact tests for categorical data. Our dataset contained minimal missing values, less than 5% across all variables, which were handled through pairwise deletion. The hypothesis tests conducted were exploratory, aimed at generating rather than confirming hypotheses; therefore, no adjustments for multiple comparisons were made. Logistic regression analyses were used to investigate potential predictors of RA development, with results presented as odds ratios and 95% confidence intervals. The agreement between the initial physician assessments and final diagnoses was measured using Cohen’s kappa coefficient. The diagnostic accuracy of ultrasonography was evaluated through the sensitivity, specificity, and positive and negative predictive value, each reported with 95% confidence intervals. We assumed no unmeasured confounding for the validity of our results, addressing potential confounders via regression-based covariate adjustment. Statistical significance was determined at a *p*-value of less than 0.05.

## 3. Results

### 3.1. Study Cohort and RA Development

Of the 503 patients screened for eligibility, 326 (65% of those screened) were included in the final analysis, with a follow-up period of 24 months. The cohort predominantly comprised females (83%), with an average age of 45.10 ± 11.37 years. During the follow-up, 123 patients (37.7% of the cohort) were diagnosed with clinical RA according to the 2010 ACR-EULAR criteria.

### 3.2. Baseline Characteristics

The demographic and laboratory characteristics at baseline are detailed in Table 1. The patients who developed RA (RA group) were significantly older on average (47.10 years, 95% CI: 45.32–48.88) compared to those who did not develop RA (non-RA group) (43.90 years, 95% CI: 42.49–45.31; *p* = 0.019). Significant differences were also observed in the baseline laboratory markers between the groups, with the RA group showing higher mean values of the white blood cell count (WBC; *p* = 0.012), erythrocyte sedimentation rate (ESR; *p* = 0.003), rheumatoid factor (RF; *p* < 0.0001), and anti-citrullinated protein antibodies (anti-CCP; *p* < 0.0001).

### 3.3. Baseline US Findings

Ultrasonography at baseline revealed the significantly higher prevalence of synovitis in any hand joint in the RA group compared to the non-RA group (84.6% vs. 16.3%; *p* < 0.0001). The RA group also exhibited the higher prevalence of positive PD signals (5.7% vs. 0.5%; *p* = 0.003). There were no significant differences between the groups in terms of carpal bone erosion, wrist ganglion cysts, bone hypertrophy, and IP joint erosion.

### 3.4. Predictors of RA Development

Both univariate and multivariate logistic regression analyses were conducted to identify predictors of RA development within 24 months. The univariate analysis identified several significant predictors, including age, WBC, ESR, RF, anti-CCP, and various ultrasonography findings. The multivariate analysis, incorporating variables with a *p*-value < 0.05, identified the strongest independent predictors of RA development: WBC (OR = 1.23, 95% CI = 1.02–1.50, *p* = 0.035), RF (OR = 1.08, 95% CI = 1.04–1.12, *p* ≤ 0.0001), anti-CCP (OR = 1.04, 95% CI = 1.02–1.07, *p* = 0.002), and certain ultrasonography findings, including radiocarpal synovial thickening, PIP and MCP synovitis, and wrist effusion (Table 2).

### 3.5. Machine Learning Model Performance

Applying various machine learning models to predict RA based on the baseline variables highlighted the random forest model, which comprised 80 trees, as the most effective, demonstrating an F1-score of 82.1. This model also showed high precision (82.11%) and sensitivity (82.09%). The performance overview of the different models is detailed in Table 3, with the ROC and precision–recall curves for the random forest model depicted in Figure 5.

### 3.6. Feature Importance in Prediction Models

The results of this analysis indicated that radiocarpal synovial thickening, MCP synovitis, PIP synovitis, RF, and anti-CCP were the most influential features, with relative importance scores of 0.23, 0.18, 0.16, 0.13, and 0.11, respectively. For a visual representation of these feature importance scores, please refer to Figure 6.

### 3.7. Impact of Ultrasonographic Features on Model Performance

Incorporating US features significantly enhanced the performance metrics across all models, particularly for the random forest model, which saw its F1-score increase from 77.8 to 82.1, precision from 77.7 to 82.11, and sensitivity from 77.9 to 82.09. Detailed improvements are documented in Table 3, Table 4 and Table 5.

## 4. Discussion

This study aimed to evaluate the prognostic value of hand US in the early detection of RA in patients with nontraumatic hand joint pain. We identified specific US features, including radiocarpal synovial thickening, PIP and MCP synovitis, and wrist effusion, as strong predictors of RA development when combined with elevated RF and anti-CCP titers.

At the conclusion of the follow-up, 37.7% of the patients had developed clinical arthritis, with 84.6% showing ultrasound synovitis at baseline, indicated by a grade of synovial hypertrophy in GS greater than 1 or the presence of PD abnormalities. Of these, 48.8% had synovitis in only one joint, while 35.9% had it in two or more joints. Clinically, patients who progressed to RA were more likely to exhibit elevated WBC counts, ESR levels, and RF and-CCP titers. US synovitis strongly correlated with the development of clinical RA at the patient level. However, positive PD abnormalities were rare (2.5%) and did not predict future clinical RA. These results align with previous research indicating that US synovitis is a significant marker in pre-RA patients and can predict the onset of clinical arthritis in those at risk [4,5,11,32,33]. However, the prevalence and predictors of RA development may vary depending on the US method, scanning protocol, population characteristics, assessment of effusion and erosion, or examination of other modes, such as PD or GS.

In a cohort study by Nam et al., patients experiencing arthralgia, irrespective of their antibody status, were followed for one year. Baseline ultrasound synovitis was defined as GS ≥ 2 and/or PD ≥ 1. By the end of the study, 16% of participants developed arthritis, 59% of whom had positive US findings at the start. The wrists (26%) and MTP joints (11%) were most frequently affected. Independent predictors of inflammatory arthritis development included age, US synovitis, a positive PD signal, and morning stiffness. Interestingly, US effectively ruled out IA in patients without US synovitis [5]. Van Beers-Tas et al. observed that among 163 seropositive individuals, 31% developed clinical arthritis after a median of 12 months, with 86% meeting the 2010 ACR/EULAR criteria for RA. Their analysis suggested that, excluding the MTP joints, the US synovial thickness correlated with both the occurrence and timing of clinical arthritis at the patient level. Positive PD indicators, as in our findings, were infrequent among those at risk and were not predictive [11]. Naredo et al. conducted a prospective study on anti-CCP+ patients without clinical arthritis, assessing GS, PD, and erosions across 32 joints, including the wrists, MCPs, PIPs, and MTPs. Eighty-six percent of those who developed clinical arthritis had at least one US abnormality at baseline, compared to 67% of non-progressors. All US findings predicted IA progression, with the highest risk associated with PD abnormalities [34]. Van Steenbergen et al. studied 192 arthralgia patients with positive autoantibody titers, finding that 23% progressed to RA after an average of 11 months. Their study indicated a trend towards US abnormalities in ≥1 joints at baseline being predictive of arthritis development. However, contrary to our findings, they demonstrated that US predicted clinical synovitis at the joint level rather than the patient level [33].

The collective evidence presented in this study suggests that the utilization of US in the assessment and stratification of the RA risk appears to be of substantial value during the preclinical stages of the disease, surpassing the sole reliance on clinical examination and autoantibody status. However, it is crucial to acknowledge that the observed inconsistencies among the findings may be attributed to the variability in the scanning protocol employed, which includes the application of US examinations solely to symptomatic joints or to all joints, encompassing the MTP joints and larger joints. Additionally, another potential explanation for the discrepancies observed in the results could be the existence of technical disparities between US machines, as these variations have a notable impact on the detection of PD signals [11].

Previous research indicates that clinicians’ requirements for US scanning vary by indication, with a higher number of joints assessed for diagnostic purposes to detect inflammation and initiate treatment promptly in cases of widespread pain. For ongoing synovial activity, the choice of joints to scan often depends on the presence of pain or the tendency of certain joints to be more frequently affected. Studies also reveal that rheumatologists in university settings typically perform US on fewer than 10 joints, constrained by the available time [35]. Although simplified US joint count scoring systems have been evaluated for early RA and found feasible in assessing joint inflammation, selecting the optimal subset of asymptomatic joints for clinical use remains challenging. Employing a multiple-joint scoring system is more time-consuming, but focusing on the joints commonly affected by RA could save time and possibly enhance the sensitivity [33]. To refine joint selection in US protocols, we analyzed all hand joints using US and employed regression models and machine learning techniques to identify the most significant US predictors of RA progression [11].

The results from a multivariate regression model and machine learning analysis indicate that specific US findings significantly influence the likelihood of RA progression in the U.S. population. Notably, radiocarpal synovial thickening, PIP and MCP synovitis, and wrist effusion at baseline, as identified by grayscale US, are strong independent predictors of RA in at-risk patients. These US variables, coupled with high RF and anti-CCP titers in patients presenting with arthralgia but not fulfilling the RA clinical criteria, are highly valuable for risk stratification in clinical settings. Additionally, the study underscores the potential of AI, particularly the random forest algorithm, in enhancing CAD systems, showing superior performance over other models.

Based on the findings, a simplified US protocol targeting the key joints and scores may enhance the efficiency and feasibility of screening and diagnosing pre-rheumatoid arthritis (pre-RA) patients. Additionally, incorporating machine learning models could improve the accuracy and objectivity in predicting RA development by using ultrasound data along with other clinical and laboratory variables. However, further research is needed to validate and optimize these US protocols and machine learning models across different settings and populations.

Another noteworthy observation is that wrist joint synovial thickening and synovitis in the PIP and DIP joints can manifest in the early stages of OA, as evidenced by prior studies [36]. Recent imaging techniques have revealed that synovial inflammation is common in both the early and late phases of OA [36], particularly affecting the DIP, PIP, and the base of the thumb [37]. In our study, the presence of US-positive findings in these joints at baseline, without the subsequent development of RA during follow-up, may have represented early manifestations of OA. However, additional research with longer follow-up is necessary to confirm this.

This suggests that US may not always be specific enough to distinguish between RA and OA, particularly when clinical and laboratory indicators are unclear. Thus, the careful interpretation of US results is essential to avoid diagnostic errors.

### Study Strengths and Limitations

This study exhibits several notable strengths. Firstly, its prospective design, coupled with a substantial sample size and an extended 24-month follow-up period, lends credibility to the findings. Additionally, the implementation of a standardized US scanning protocol enhanced the reliability and reproducibility of the data collection process. Furthermore, the innovative application of machine learning techniques facilitated the identification of the most predictive US features, underscoring the study’s methodological rigor.

However, it is essential to acknowledge certain limitations. As a single-center study, the generalizability of the findings may be restricted, necessitating further validation across diverse healthcare settings and populations. Despite concerted efforts to adjust for key variables, the potential for residual confounding cannot be entirely ruled out. Moreover, while the US protocol employed in this study was comprehensive, there may be alternative joint areas or scoring systems that could yield additional predictive value, warranting further exploration.

Consequently, future research endeavors spanning multiple centers, diverse populations, and the exploration of alternative US protocols or scoring systems are recommended. Such initiatives would not only validate the present findings but also refine and enhance the predictive capabilities of the proposed approach. By addressing these limitations, the study’s impact could be amplified, contributing to the development of more robust and generalizable predictive models for the early detection of rheumatoid arthritis.

## 5. Conclusions

This prospective cohort study exhibits the significance of targeted hand ultrasonography in the identification of individuals with a strong likelihood of advancing to rheumatoid arthritis. Key ultrasonographic characteristics, such as thickening of the radiocarpal synovium and synovitis in the PIP and MCP joints, as well as the presence of fluid in the wrist, were found to be strong and independent predictors of the onset of clinical rheumatoid arthritis within a 2-year period of follow-up among patients displaying undifferentiated arthritis. Moreover, the machine learning analysis highlighted these ultrasound predictors, in addition to increased levels of RF and anti-CCP antibodies, as the most influential differentiating factors when compared to other variables.

## Figures and Tables

**Figure 1 diagnostics-14-01181-f001:**
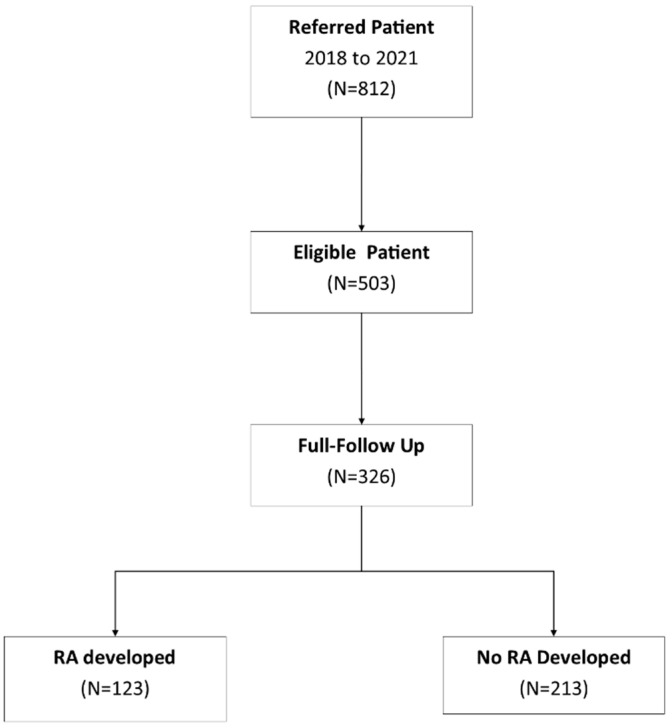
Study flow chart.

**Figure 2 diagnostics-14-01181-f002:**
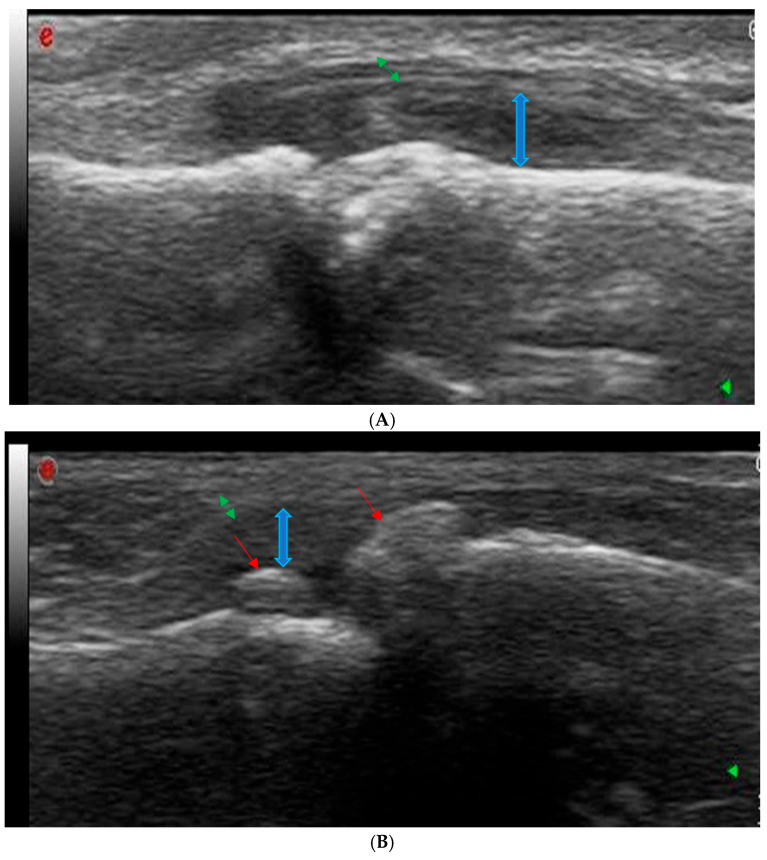
Examples of joint gray scale ultrasound findings. (**A**) Wrist joint showing synovial hypertrophy (indicated by double-headed blue arrows) and joint capsule distention (double-headed green arrows). (**B**) Proximal interphalangeal (PIP) joint showing synovial hypertrophy (indicated by double-headed blue arrows) and joint capsule distention (double-headed green arrows). Osteophytes are marked by red arrows.

**Figure 3 diagnostics-14-01181-f003:**
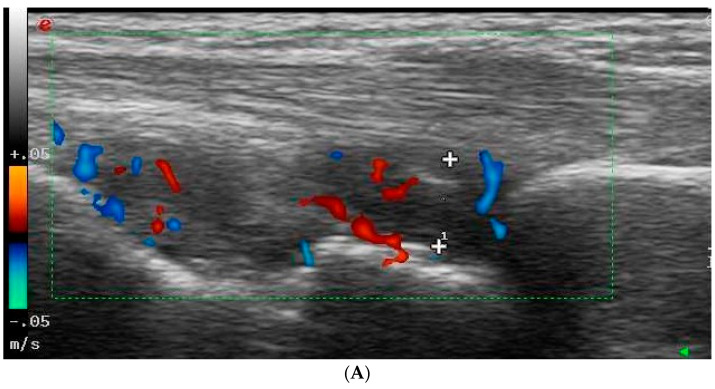
Examples of joint Doppler ultrasound findings. (**A**) Radiocarpal joint showing synovial thickening, marked as “1” within the green rectangle, and increased vascularity on Doppler ultrasound, indicated by blue and red areas within the green rectangle. (**B**) Proximal interphalangeal (PIP) joint with synovitis and hypervascularity on Doppler ultrasound, indicated by blue and red areas within the green rectangle.

**Figure 4 diagnostics-14-01181-f004:**
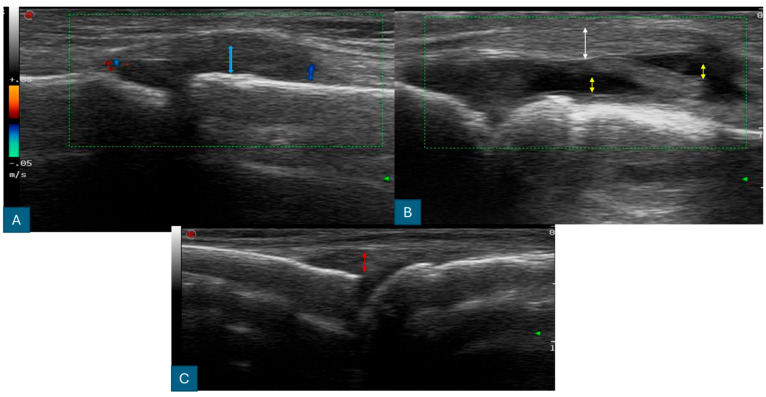
Ultrasound findings of the MCP joint. (**A**) Grade II joint capsule distension is observed without significant hypervascularity. The double-headed blue arrows indicate synovial thickness. (**B**) Moderate joint effusion and joint capsule thickening. The double-headed white arrows indicate joint capsule thickness, yellow arrows show joint effusion, and the green rectangle encompasses Color Doppler imaging to show vascularity. (**C**) A normal MCP joint with standard synovial thickness, as shown by the double-headed blue arrows. The red double-headed arrow represents the normal MCP joint anatomy.

**Figure 5 diagnostics-14-01181-f005:**
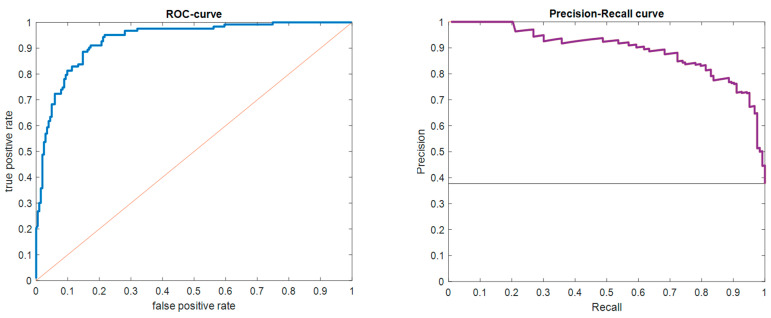
The ROC (**left panel**) and precession–recall (**right panel**) curves.

**Figure 6 diagnostics-14-01181-f006:**
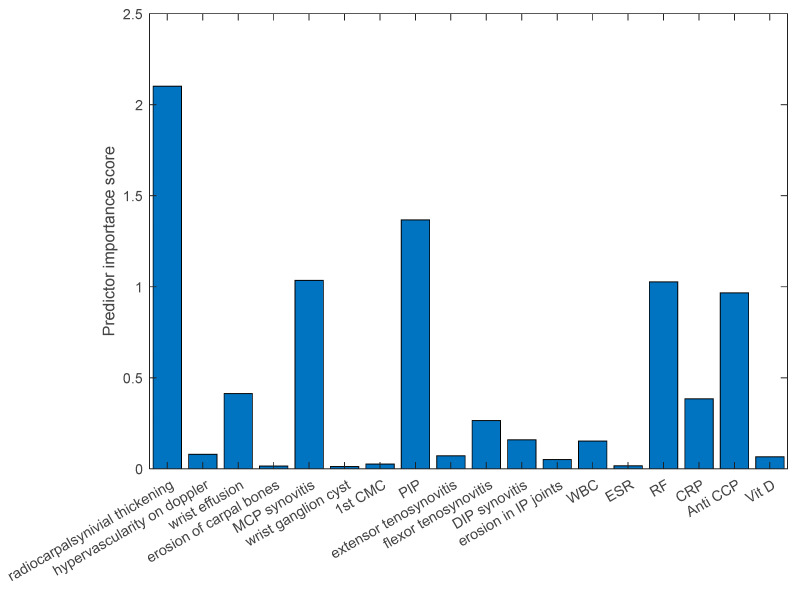
The predictive value of the US and laboratory variables according to the random forest algorithm.

**Table 1 diagnostics-14-01181-t001:** Baseline demographic and laboratory characteristics.

Variable	Non-RA Patients (*n* = 203)	RA Patients (*n* = 123)	*p*-Value
Age (year) (mean ± SD)	43.90 ± 10.38	47.10 ± 12.65	0.019
Gender, female, *n* (%)	174 (85.7)	97 (78.9)	0.109
WBC	6.81 ± 2.17	7.45 ± 2.18	0.012
ESR	17.13 ± 15.26	23.11 ± 20.12	0.003
RF	7.13 ± 5.68	21.63 ± 29.57	<0.0001
CRP	5.15 ± 10.76	7.32 ± 9.70	0.069
Anti-CCP	3.93 ± 9.14	30.95 ± 75.14	<0.0001
Vitamin D	33.64 ± 15.72	34.93 ± 18.65	0.504

**Table 2 diagnostics-14-01181-t002:** Independent predictive variables associated with RA progression for 24-month follow-up.

Variable	Univariate Model		Multivariate Model	
	Odds Ratio (95%CI)	*p*-Value	Odds Ratio (95%CI)	*p*-Value
Age (years)	1.03 (1.01–1.05)	0.015	1.03 (0.99–1.07)	0.147
WBC	1.14 (1.03–1.27)	0.013	1.23 (1.02–1.50)	0.035
ESR	1.02 (1.006–1.03)	0.004	1 (0.98–1.03)	0.804
RF	1.09 (1.05–1.12)	<0.0001	1.08 (1.04–1.12)	<0.0001
Anti-CCP	1.05 (1.02–1.08)	0.003	1.04 (1.02–1.07)	0.002
Radiocarpal–synovial thickening	18.19 (10.18–32.50)	<0.0001	39.87 (15.86–100.20)	<0.0001
Hypervascularity	12.19 (1.48–100.31)	0.02	7.99 (0.32–202.95)	0.208
Wrist effusion	4.68 (1.63–13.47)	0.004	12.56 (2.22–70.95)	0.004
MCP synovitis	51.53 (6.88–385.86)	<0.0001	39 (3.34–454.1)	0.003
PIP synovitis	28.68 (8.61–95.55)	<0.0001	68 (12.62–365.91)	<0.0001
Extensor tenosynovitis	3.89 (1.54–9.83)	0.004	4.03 (0.60–27.01)	0.151
Flexor tenosynovitis	2.77 (1.16–6.61)	0.022	0.74 (0.14–3.96)	0.724
DIP synovitis	4.40 (1.35–14.36)	0.014	0.76 (0.10–6.16)	0.8

**Table 3 diagnostics-14-01181-t003:** The results of the evaluation metrics for machine learning methods including tree, linear SVM, quadratic SVM, SVM with Gaussian kernel, KNN, AdaBoost, random forest, and NN for US and laboratory features.

US and Laboratory Features	Tree	Linear SVM	Quadradic SVM	Gaussian SVM	KNN K = 10	AdaBoost	Random Forest	Neural Network
Accuracy	83. 7	84.97	84.66	84.97	76.07	86.2	86.5	85.28
Sensitivity	78.05	78.86	73.98	78.05	42.28	79.67	82.11	79.67
Specificity	87.19	88.67	91.13	89.16	96.55	90.15	89.16	88.67
Precision	78.69	80.83	83.49	81.36	88.14	83.05	82.09	80.99
FPR	12.81	11.33	08.87	10.84	03.45	09.85	10.84	11.33
F1-Score	78.37	79.84	78.45	79.67	57.14	81.33	82.10	80.33
MCC Matthews	65.35	67.87	66.90	67.79	48.88	70.42	71.28	68.57
Kappa Cohen’s	65.35	67.86	66.61	67.75	43.26	70.39	71.28	68.56

**Table 4 diagnostics-14-01181-t004:** The results of the evaluation metrics for machine learning methods including tree, linear SVM, quadratic SVM, SVM with Gaussian kernel, KNN, AdaBoost, random forest, and NN for US features.

US Features	Tree	Linear SVM	Quadradic SVM	Gaussian SVM	KNN K = 10	AdaBoost	Random Forest	Neural Network
Accuracy	82.52	82.52	81.90	80.98	78.83	83.44	83.74	81.60
Sensitivity	69.92	78.05	69.92	73.98	47.15	68.29	75.61	65.04
Specificity	90.15	85.22	89.16	85.22	98.03	92.61	88.67	91.63
Precision	81.13	76.19	79.63	75.21	93.55	84.85	80.17	82.47
FPR	09.85	14.78	10.84	14.78	1.97	7.39	11.33	8.37
F1-Score	75.11	77.11	74.46	74.59	62.7	75.68	77.82	72.73
MCC Matthews	62.15	62.98	60.84	59.4	55.81	64.20	65.08	60.08
Kappa Cohen’s	61.75	62.97	60.54	59.4	50.08	63.34	65.01	59.19

**Table 5 diagnostics-14-01181-t005:** The results of the evaluation metrics for machine learning methods including tree, linear SVM, quadratic SVM, SVM with Gaussian kernel, KNN, AdaBoost, random forest, and NN, for laboratory features.

Laboratory Features	Tree	Linear SVM	Quadradic SVM	Gaussian SVM	KNN K = 10	AdaBoost	Random Forest	Neural Network
Accuracy	63.8	72.09	71.47	70.55	70.55	70.55	73.01	65.34
Sensitivity	54.47	30.89	34.15	30.89	30.89	42.28	48.78	49.59
Specificity	69.64	97.04	94.09	94.58	94.58	87.68	87.68	74.88
Precision	51.94	86.36	77.78	77.55	77.55	67.53	70.59	54.46
FPR	30.54	02.96	05.91	05.42	05.42	12.32	12.32	25.12
F1-Score	53.17	45.51	47.46	44.19	44.19	52	57.69	51.91
MCC Matthews	23.72	39.63	36.81	34.55	34.55	34.19	40.26	24.98
Kappa Cohen’s	23.70	31.99	31.74	28.90	28.90	32.34	38.83	24.91

## Data Availability

The datasets generated and analyzed during the current study are available from the corresponding author on reasonable request.

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
