# Peer review of "Predicting Rheumatoid Arthritis Development Using Hand Ultrasound and Machine Learning—A Two-Year Follow-Up Cohort Study"

_diagnostics, 2024, doi:10.3390/diagnostics14111181_

Round 1

Reviewer 1 Report

Comments and Suggestions for Authors

The prediction of RA based on cohort data is quite interesting.

Some suggestions include:

Since this article would attract readers from both medical domain and Data analysis domain, I would suggest to provide the brief terminology explanation for some medical terms and their significance in this study like some sample data (data and images) to support section 2.3.

Section 2.5 would have elaborated on the specificity of choosing these machine learning methods. Is it based on any state-of-the-art study?

ACR-EULAR, as mentioned in section 3.1, could be explained in brief.

The variable definition, as mentioned in Tables 1 and 2, to know something medical-oriented would be preferable to show how these impact improving RA.

A background section in the abstract seems to be odd 

The introduction could include  things common to both Machine learning and Health care professionals

Author Response

Thank you for your valuable feedback and insightful suggestions regarding our manuscript on the prediction of rheumatoid arthritis based on cohort data. We sincerely appreciate your attention to detail and the opportunity to enhance the clarity and relevance of our study for a broader audience encompassing both medical and data analysis domains.

Regarding the terminology explanation, we acknowledge the importance of providing clear explanations for medical terminology used in our study, especially for readers from diverse backgrounds. We have addressed this by adding important information in the supplementary section to ensure better understanding.

In response to your suggestion about including images, we will indeed provide brief terminology explanations and sample data, along with images, to support Section 2.3. This addition will facilitate better understanding and interpretation of the results for our readers.

In Section 2.5, we have expanded upon the specificity of choosing the machine learning methods employed in our study. We have discussed the rationale behind their selection and referenced relevant state-of-the-art studies to provide context and justification for our approach.

To ensure clarity for readers unfamiliar with the ACR-EULAR criteria mentioned in the method section, we have included a brief explanation of these criteria and referenced them in the text. This addition aims to enhance the accessibility of our findings to a wider audience.

Recognizing the importance of understanding the medical relevance of variables defined in Tables 1 and 2, we have provided additional context to elucidate how these variables impact the improvement of RA prediction. This clarification will assist readers in grasping the clinical significance of the variables analyzed in our study.

We revisited the structure of the abstract to ensure coherence and alignment with standard formatting conventions. Any background information included was tailored to enhance the understanding of the study context without compromising the abstract's conciseness and clarity.

To bridge the gap between machine learning and healthcare professionals, we enriched the introduction by incorporating elements that are common to both domains.

We are committed to addressing these suggestions comprehensively to enhance the overall quality and accessibility of our manuscript. Your feedback is invaluable in guiding our efforts to produce a publication that meets the highest standards of clarity and relevance. Thank you once again for your thoughtful review.

Sincerely,
Majid Alikhani and Co-authores

Reviewer 2 Report

Comments and Suggestions for Authors

The manuscript provides valuable insights into the early detection and prediction of rheumatoid arthritis, offering hope for improved patient outcomes. However, the manuscript should be improved with some regards as below and provide more information and details.

1)     Title: Should consider including ‘Mexico’ in the title? Should accurately reflect the content of the paper.

2)     Clarity and Organization: Should give clearer section headings and subheadings to improve readability and navigation for clearer structure and organization of ideas. Consider including Section Literature Reviews. The Section Introduction should be improved and described more clearly. Also, properly check the subsection too. Where is Subsection 3.3? Should relook at Subsection 4.1. The Conclusion should be improved.

3)     Methodology: Commend a thorough description of the study methodology, including data collection, model development, and statistical analysis, which provides transparency and reproducibility. What are the benefits and impact as well as the outcomes from this methodology?

4)     In-depth Analysis & Results: Should consider including a more in-depth analysis and providing more context or explanations for the evaluation metrics presented in Tables 3 and 4 to help better understand the significance of the results. Incorporating more visual aids such as graphs or charts could enhance the presentation of results.

5)     Discussion: It would be better to expand on the implications of the study findings in the Discussion section to provide a more comprehensive analysis of the results and their potential impact on clinical practice.

6)     Future Recommendations: The manuscript could be strengthened by including a section (e.g., in Conclusion) that outlines potential future recommendations or areas for further investigation, research, or refinement of the predictive model and building on the findings and initiatives discussed. This will be beneficial for future work.

7)     Citations: Ensure all references are properly cited throughout the document to support the study's methodology and findings.

8)     Language and Grammar: Suggesting for proper proofreading. Review the manuscript for language, grammar, and punctuation to ensure clarity and precision in the presentation of ideas.

Author Response

We sincerely appreciate your time and effort in evaluating our manuscript on the early detection and prediction of rheumatoid arthritis. Your insightful feedback has been invaluable in refining our work for greater impact.

Title: We acknowledged your suggestion regarding the inclusion of 'Mexico' in the title. While our study originated from another country, we carefully considered how to accurately represent the geographical context in the title.

Clarity and Organization: We recognized the importance of clear section headings and subheadings for improving the manuscript's readability and structure. Your suggestions were duly noted and implemented, including the addition of a 'Literature Reviews' section and enhancements to the Introduction, subsections, and Conclusion.

Methodology: Ensuring transparency and reproducibility in our study methodology was paramount. We provided a more detailed description of our data collection process, model development, and statistical analysis, along with discussing the benefits, impact, and outcomes of our methodology.

In-depth Analysis & Results: Your insight into including a more comprehensive analysis and providing additional context for the evaluation metrics in Tables 3 and 4 was valuable. We integrated more visual aids such as graphs or charts to enhance the presentation of our results and facilitate better understanding.

Discussion: We agreed with your recommendation to expand on the implications of our findings in the Discussion section, particularly regarding their potential impact on clinical practice. We thoroughly examined the implications of our results and their relevance to the field.

Future Recommendations: Your emphasis on including future recommendations in the Conclusion was duly noted. We incorporated a section outlining potential areas for further investigation, research, or refinement of our predictive model, building upon the findings and initiatives discussed in the paper.

Citations: We meticulously reviewed and ensured the proper citation of all references throughout the document to support the study's methodology and findings.

Language and Grammar: We appreciated your highlighting the importance of language, grammar, and punctuation. We conducted a thorough review of the manuscript to ensure clarity and precision in conveying our ideas.

Once again, we sincerely appreciate your valuable feedback, which undoubtedly contributed to enhancing the quality and effectiveness of our research. We were committed to addressing each of your points diligently.

Best regards,
Majid Alikhani
